# Exploring Nurses’ Quit Intentions: A Structural Equation Modelling and Mediation Analysis Based on the JD-R and Social Exchange Theories

**DOI:** 10.3390/healthcare13070692

**Published:** 2025-03-21

**Authors:** Dhurata Ivziku, Blerina Duka, Alketa Dervishi, Lucia Filomeno, Fabrizio Polverini, Ippolito Notarnicola, Alessandro Stievano, Gennaro Rocco, Cesar Ivan Aviles Gonzalez, Nertila Podgorica, Daniela D’Angelo, Anna De Benedictis, Francesco De Micco, Raffaella Gualandi, Marzia Lommi, Daniela Tartaglini

**Affiliations:** 1Department of Health Professions, Fondazione Policlinico Universitario Campus Bio-Medico, 00128 Rome, Italy; d.tartaglini@policlinicocampus.it; 2Faculty of Medicine, University “Our Lady of the Good Counsel”, 1000 Tirana, Albania; bleriduka@yahoo.it; 3Faculty of Medicine, Western Balkans University, 1000 Tirana, Albania; alketa.dervishi@wbu.edu.al; 4Department of Clinical and Molecular Medicine, University La Sapienza, 00189 Rome, Italy; 5Department of Strategic Directory, Local Health Authority 1 Liguria, 18038 Bussana di Sanremo, Italy; polve63@gmail.com; 6Department of Medicine and Surgery, University of Kore, Piazza dell’Università, 94100 Enna, Italy; ippo66@live.com (I.N.); cesar.aviles@unikore.it (C.I.A.G.); 7Department of Clinical and Experimental Medicine, University of Messina, 98122 Messina, Italy; alessandro.stievano@gmail.com; 8International Center for Nursing Research Montianum Our Lady, Good Counsel Catholic University, 1000 Tirana, Albania; genna.rocco@gmail.com; 9Department of Nursing, Health University of Applied Sciences Tyrol, 6020 Innsbruck, Austria; nertila.podgorica@fhg-tirol.ac.at; 10Department of Nursing Science and Gerontology, UMIT–Private University for Health Sciences, Medical Informatics and Technology, 6060 Hall in Tirol, Austria; 11Department of Health Professions, Local Health Authority Rome 6, 00041 Albano Laziale, Italy; daniela.dangelo@aslroma6.it; 12Research Unit in Nursing Science, Department of medicine and surgery, University Campus Bio-Medico, 00128 Rome, Italy; a.debenedictis@policlinicocampus.it (A.D.B.); r.gualandi@unicampus.it (R.G.); 13Department of Clinical Affair, Fondazione Policlinico Universitario Campus Bio-Medico, 00128 Roma, Italy; f.demicco@policlinicocampus.it; 14Research Unit of Bioethics and Humanities, Department of Medicine and Surgery, Università Campus Bio-Medico di Roma, 00128 Roma, Italy; 15Department of Nursing, University La Sapienza, 00157 Rome, Italy; marzia.lommi@gmail.com

**Keywords:** nurses, quit intention, retention, mediation, structural equation modelling, workload, autonomy, LMX, social exchange theory, job demands–resources

## Abstract

Background/Objectives: Understanding how work environments affect nurses’ turnover intentions is crucial for improving retention and organizational stability. Research on factors influencing nursing turnover intentions in Italy is limited despite its significant impact on healthcare sustainability today. Aim: This study aims to identify the individual, interpersonal, and job-related factors influencing quit intentions among nurses, examining the mediating role of job satisfaction. Methods: Guided by the Job Demands–Resources (JD-R) theory and Social Exchange Theory (SET), this cross-sectional study applied structural equation modelling (SEM) to analyse theoretical relationships. Researchers collected data between August 2022 and December 2023 via an online survey sent to nurses in different settings across Italy. This study tested a hypothesised mediation model using SEM analysis, demonstrating good fit indices. Results: A total of 1745 nurses responded. The findings reveal that high job demands—physical, mental, and emotional workloads—were significantly associated with increased dissatisfaction and quit intentions. Conversely, job resources, including decision-making autonomy, inspiring leadership, and positive leader–member exchanges, were linked to greater job satisfaction and retention. Contrary to expectations, work satisfaction did not mediate the relationship between job demands/resources and turnover intentions. This could be explained by the strong direct effect of job satisfaction on turnover intentions. Additionally, younger nurses were more likely to report higher turnover intentions. Conclusions: Identifying potential quitters at an early stage is essential for the sustainability of healthcare organizations. Understanding the factors contributing to nurse turnover is crucial for developing effective retention strategies. This study emphasizes the need for healthcare organizations to prioritize supportive work environments to enhance nurse job resources, well-being, and retention.

## 1. Introduction

The global nursing shortage poses a significant challenge to healthcare systems and the quality and safety of care coverage. Nurse turnover rates are a critical factor in exacerbating this issue. International data report 4% to 54% nurse turnover rates [1,2]. For every nurse that leaves the job, the experience of the nurses that stay gets negatively affected, with additional job pressure and workload, pushing more nurses to quit the profession [3,4]. This combination of high turnover and staffing shortages threatens the sustainability of the global nursing workforce [3]. Understanding the factors contributing to nurse turnover is crucial for developing effective retention strategies [5,6]. Addressing these areas is vital for ensuring nurse well-being, patient safety, the overall quality of care, and the sustainability of healthcare systems.

For several decades, nurses’ retention and turnover intentions have been a significant concern in healthcare research. This issue has gained renewed urgency during and after the COVID-19 pandemic [2,3]. In Italy, the nursing workforce faces critical challenges, with the nurse-to-population ratio and nursing graduation rates falling below the European Community average [4]. Previous research conducted in Italy during the early 21st century revealed high turnover intentions among nurses. Some studies reported that 35.5% of nurses expressed an intention to leave their current position [5], and 34.4% intended to leave the workplace within the first year of employment in the setting [6]. Furthermore, a significant proportion of nurses (33.1%) also indicated an intention to leave the nursing profession [5]. In addition, a parallel issue is the number of nursing students withdrawing from their education programs. According to some studies [7], this rate reaches 33% of the nursing student population, creating a critical gap in the nursing pipeline. Therefore, promoting positive clinical experiences and reducing the risk of program withdrawal is essential to ensuring a sustainable supply of qualified nurses for the future [8,9]. The country faced additional challenges, with a 3% increase in resignations among working-age nurses and 55,534 nurses’ retirement from 2021–2022 [10]. These factors have exacerbated the workforce deficit, further intensifying the staffing crisis. This shortage indicates a healthcare system under increasing strain to meet escalating care demands.

The COVID-19 outbreak led to significant nurse staffing challenges and increased turnover rates worldwide. Understanding, measuring, and addressing nurses’ turnover intentions is crucial. This should be addressed at a country level, as this phenomenon is highly sensitive to each country’s specific context and conditions [11]. Therefore, examining the causes and determinants of nurses’ intentions to leave in the post-COVID-19 period in Italy is essential. Consequently, this research team aims to enhance the body of knowledge specific to the nursing workforce in Italy while also contributing to the international field.

Understanding the reasons for turnover is imperative for improving retention strategies. Prioritizing the implementation of retention policies and identifying best practices to enhance healthcare workers’ resilience is crucial [12]. A comprehensive understanding of the work environment and the dynamic relationships among individual variables, interpersonal relations, and job characteristics can provide useful information to address nurses’ quit intentions [13,14]. Recognizing the factors that increase or reduce turnover intentions will provide healthcare managers with specific areas of focus to effectively retain nurses and better prepare for future public health crises [12]. While acknowledging the value of these factors in informing targeted retention strategies by healthcare managers, the existing body of research in this area remains incomplete.

Despite the extensive research conducted on turnover intentions in nursing, there are still significant gaps in the literature, particularly with regard to the evolving dynamics of workforce retention in the post-COVID-19 era. Existing research has primarily focused on general turnover trends or has been conducted in different healthcare systems, often overlooking the unique structural, cultural, and economic factors shaping nurse retention from a national point of view, especially in Italy. Moreover, while studies have identified various determinants of turnover, the interplay between job demands, job resources, and job satisfaction remains insufficiently explored, particularly in the context of the ongoing nursing shortage. Some studies suggest that job satisfaction mediates the relationship between job demands/resources and turnover intentions, whereas others indicate that it plays an independent predictive role, highlighting an inconsistency that warrants further examination. Additionally, previous research has frequently relied on pre-pandemic data, which hinders the assessment of whether the determinants of nurse turnover have been altered in response to the unprecedented challenges posed by the pandemic. Addressing these critical gaps, this study provides a necessary update and contributes to a more nuanced understanding of nurse retention by applying a robust theoretical framework to an underexplored national context.

Therefore, this study answers the following research questions: What factors reduce or increase Italian nurses’ intentions to leave? Does job satisfaction mediate these relationships?

The primary aim of this study is to identify the individual and interpersonal interactions and job-related factors influencing intentions to leave among Italian nurses. Additionally, this study seeks to examine the mediating role of job satisfaction in these associations.

### 1.1. Theoretical Frameworks

Psychological and behavioural science supports the idea that expressed intentions can serve as reliable predictors of subsequent actions [7]. Therefore, the intention to leave a job can be a significant indicator of actual turnover behaviour. It represents an employee’s behavioural intention or attitude towards departing from an organization or unit [8]. The intention to leave highlights job-related dissatisfaction or disengagement and serves as an early warning signal to prevent turnover [9]. By comprehending employees’ intentions, organizations can proactively address and mitigate potential turnover before it occurs.

The intention to leave is a complex and multidimensional process. About 36 factors influence nurses’ turnover intentions [2,10]. These factors are grouped into four broad categories: *individual* (personal characteristics, such as age, gender, marital status, education, and tenure, and psychological factors like self-efficacy, autonomy, vulnerability, and burnout); *job-related* (work context factors, such as workload, role ambiguity, stress, job satisfaction, and commitment); *interpersonal* (such as supervisor support, leadership style, recognition, relationships, empowerment, and social support); and *organisational* (such as the organisational climate, organisational structure, financial determinants, salary, and career opportunities) [11]. These factors have varying degrees of influence on nurses’ intentions to leave their positions.

During and after the COVID-19 pandemic, the healthcare workforce has shown increased vulnerability to the witnessed growing job demands [15]. This growing strain is closely linked to turnover intentions and can be better understood through the lenses of the Job Demand–Resources (JD-R) Theory [16] and Social Exchange Theory (SET) [17].

JD-R describes work experiences as shaped by two opposing forces: job demands and job resources. Job demands encompass those physical, psychological, social, or organizational components of the job that necessitate persistent physical and/or psychological (cognitive and emotional) effort. While job demands are not inherently harmful, they can become stressors when they require sustained effort and drain energy. Job resources include support systems like well-organized environments, decision-making autonomy, recognition, and growth opportunities. These resources generate motivation; help employees manage job demands; enhance well-being; and promote engagement, commitment, and job satisfaction.

The JD-R theory highlights the importance of mapping job demands and resources as a diagnostic tool to determine if a job imposes excessive demands or lacks sufficient resources. This balance is crucial, as job demands and resources trigger two distinct processes: the health-impairment process and the motivational process [18]. In the health-impairment process, excessive demands deplete employees’ cognitive and physical resources, leading to exhaustion and burnout. Conversely, in the motivational process, adequate job resources function as intrinsic motivators that promote growth and development or as extrinsic motivators that support goal achievement, enhancing performance.

The JD-R theory emphasizes that jobs are most effective when demands are manageable and resources are adequate. In any workplace, it is crucial to consider not only job demands and resources individually but also the interaction between them alongside specific job characteristics.

SET [17] describes workplace relationships as reciprocal exchanges. It encompasses two types of social exchanges: perceived organizational support, which considers the employee–organization relationship, and the leader–member exchange, which considers supervisory–employee interactions. This study focuses on leader–member exchanges.

Exchanges can be either visible through direct actions or less visible, as inactive or psychological exchanges. Positive actions by leaders or teams, such as promoting justice and providing organizational support, encourage positive actions/reactions among employees. In contrast, negative actions, such as incivility, abusive supervision, and bullying, lead to negative reactions [19].

Psychological (inactive) exchanges form the foundation of relationships between individuals or groups and are more difficult to detect. Inactive exchanges involve withholding desirable or undesirable behaviours in the workplace. Positive psychological exchanges foster positive relationships, while negative psychological exchanges lead to negative associations. These exchanges need careful consideration by leaders because they can be particularly damaging to institutions, as inactive behaviours are more likely to be destructive than constructive [15].

Key factors like meaningful work, strong interpersonal connections, and high-quality leader–member exchanges significantly influence employee satisfaction and productivity, creating a more committed workforce. Employees who feel valued, supported, and connected are more likely to exhibit higher engagement, better performance, and reduced turnover intentions. From the perspective of the JD-R, these factors can be interpreted as job resources.

### 1.2. Study Model

This study explores individual and interpersonal interactions and job-related factors that are associated with nurses’ turnover intentions [20]. Figure 1 presents a visualization of the model tested in this study.

The *job-related* factors explored are physical, emotional, and mental workloads. These variables are classified as job demands in the JD-R theory.

The *interpersonal* factors included in this study are the inspiration of the leader and the quality of the leader–member exchange relationship. According to SET, these aspects are considered job resources in the JD-R theory and are essential to building positive relationships at work.

The *individual* factors considered in the model are nurses’ autonomy and age. These are seen as personal resources in the JD-R theory. In this study, we included nurses’ autonomy as a job resource and the nurses’ age as a control variable of turnover intentions.

The *dependent* variable in the study model is nurses’ intentions to leave the unit, institution, or profession. Turnover intention is often regarded as an outcome variable in the JD-R theory. Similarly, job satisfaction, considered a mediator in this study model, is commonly reported as either an outcome or a mediator in both the JD-R and SET theories.

This research team actively contributes to nursing science by advancing the understanding of issues within the nursing practice in Italy to support institutional and political decision-making. Previous research provided a deeper understanding of workloads [20,21,22,23], the relational dynamics between nurses and nurse managers [24,25,26,27], nurse manager competencies [28,29], and systemic challenges within the healthcare sector [30,31]. This study will advance knowledge and contribute to the identification of strategies to reduce nurse turnover, enhance the profession’s appeal, and retain younger nurses. By identifying barriers to effective nursing practice and workforce sustainability, this team seeks to illuminate areas for improvement and propose actionable recommendations. These insights will contribute to developing supportive work environments and cultivating a sustainable nursing workforce, which is essential for bolstering the resilience of Italy’s healthcare system.

### 1.3. Hypothesis Development

#### 1.3.1. Job Demands

According to the JD-R theory, job demands include organizational or personal factors that require physical or mental effort from an employee [16]. In the nursing literature, job demand has often been explored through the lens of work overload. Work overload focuses on quantitative job demands like time pressure or inadequate staffing resources [32]. Studies consistently show a direct correlation between excessive workloads and turnover intentions [33,34], dissatisfaction [35], and burnout [36]. Similarly, work pace and emotional and mental demands have been significantly associated with nurses’ intentions to leave [37,38,39] and have been linked to high levels of stress and burnout [40]. Furthermore, job demands are closely related to job satisfaction, which, in turn, has been positively correlated with retention [2].

This study focuses exclusively on physical, mental, and emotional workloads as job demands. The motivation for this choice lies in the aim to contribute to and expand the existing literature on nursing workloads within the country.

Building on the principles of the JD-R theory and supported by the evidence outlined in the literature, we formulated the following hypotheses:

**Hypothesis** **1.***Job demands (physical, mental, and emotional workloads) have a direct positive effect on nurses’ intentions to leave (the unit, institution, or profession)*.

**Hypothesis** **2.***Job demands (physical, mental, and emotional workloads) have a direct negative effect on nurses’ job satisfaction*.

#### 1.3.2. Job Resources

According to JD-R [16], job resources are the elements that help employees achieve their work goals, alleviate job demands, or foster personal development. Social and leadership support are key job resources. Mutual support within the workplace is reflected in exchanges, as outlined by Social Exchange Theory [17]. After COVID-19, significant changes have occurred in employee behaviour and organizations. Workplace relationships and support have become increasingly complex [2]. Therefore, in this study, we chose to explore relational aspects, such as inspiration by the leader and leader–member exchanges, as key job resources.

Leadership plays a particularly critical role in retaining nurses [41,42,43,44]. Nurse manager leadership can act as a powerful job resource by promoting work–life balance, providing emotional and instrumental support, and creating growth opportunities for the nursing staff [45,46,47]. SET reinforces this by highlighting the importance of mentorship and advocacy in fostering strong interpersonal connections between leaders and employees, thereby increasing employees’ sense of belonging and psychological safety [48]. Leader–member relationships of high quality have been associated with significant increases in nurses’ job performance, commitment, and retention [2,49,50,51,52].

Transformational leadership is widely recognized as the relational leadership style that yields superior outcomes for nurses, leaders, and organizations [2]. Research has shown that it enhances nurses’ motivation, vigour, and morale [53]. In this leadership style, nurse managers implement actions that inspire and motivate their staff [2,53]. Nurses who feel inspired by their leaders and perceive strong supervisory support, authentic management, transformational leadership, and positive interpersonal relationships tend to exhibit higher levels of engagement and job satisfaction [32].

Research has shown that positive leadership, adequate staffing, and open communication reduce turnover rates [19,20,21]. Nurses are more likely to remain in their positions when they feel valued and supported by their colleagues, supervisors, and the institution [54,55]. On the contrary, a toxic or overly stressful work environment, characterised by poor management, a lack of teamwork, or insufficient support from colleagues and supervisors, can increase nurses’ intentions to leave [2].

Nurses who feel supported by their leaders are more likely to experience greater autonomy in the workplace [2,56]. Autonomy is considered an individual job resource in the JD-R theory [16]. Higher levels of nurses’ autonomy, professional identity, and competence have been associated with better retention rates and nurse job satisfaction [2,32,57].

Building on the adopted theories and the insights gained from the literature review, it is reasonable to assume the following:

**Hypothesis** **3.***Job resources (nurses’ autonomy, inspiration by the leader, and the quality of relationship with the leader) directly negatively affect nurses’ intentions to leave (the unit, institution, or profession)*.

**Hypothesis** **4.***Job resources (nurses’ autonomy, inspiration by the leader, and the quality of relationship with the leader) have a direct positive effect on nurses’ job satisfaction*.

#### 1.3.3. Job Satisfaction

Employees’ perceptions and evaluations of their work life influence the decisions they make. Better work environments can reduce nurses’ likelihood of job dissatisfaction, burnout, and intention to leave by 28% to 32% [58]. Empirical evidence indicates a consistent inverse relationship between job satisfaction and turnover intention: as job satisfaction increases, turnover intentions decrease [7,12,28]. The evidence consistently supports that job satisfaction is crucial in the relationship between various antecedents and nurses’ intentions to leave [31,32,59].

Job satisfaction has been shown to mediate the relationship between leader–member exchanges and turnover intentions [60] and between job stress and the intention to leave [61]. More recently, a study on hospital nurses across eight European countries found that job resources and job demand influenced the intention to leave through the mediation of job satisfaction, work engagement, and burnout [59]. Conversely, another study suggests using job satisfaction as a predictor of employee willingness to stay, given the strong effect size observed in the analysis [62].

Considering the current evidence on the mediation of job satisfaction between job demands or resources and nurses’ intentions to leave, it can be concluded that the findings remain inconsistent. Therefore, additional research testing this mediation effect could be valuable for further theoretical development. In this study, with the aim of exploring these effects, we formulated the following:

**Hypothesis** **5.***Nurses’ job satisfaction has a direct negative effect on nurses’ intentions to leave*.

**Hypothesis** **6.***Nurses’ job satisfaction mediates the relationship between job demands, job resources, and the intention to leave*.

#### 1.3.4. Intention to Leave

Recognizing key direct and indirect factors influencing turnover intentions can help identify nurses at high risk, facilitating targeted interventions to minimize attrition and enhance retention.

In addition to the previously explored determinants of nurses quitting their workplace, socio-demographic factors may increase or decrease these intentions among nurses.

The findings from a recent systematic review [2] indicate that nurses’ turnover intentions increase with factors such as work tenure, extended work schedules, low wages, understaffing, direct care responsibilities, and rotating shifts. Additionally, a higher risk was identified among females and those with significant family responsibilities [2].

Numerous studies have identified a significant relationship between turnover intentions and age. Younger nurses, particularly those under 39 years old, are reported to be more likely to consider leaving both their hospital and the healthcare profession [2,63]. This finding is consistent with previous studies conducted among Italian hospital nurses [6]. Conversely, during crises, older workers have also demonstrated higher intentions to leave [2].

Considering the present debate on this aspect, with the aim to explore quitting intentions among all nurses in the post-pandemic era, this study tested the following:

**Hypothesis** **7.***Nurses’ turnover intentions decrease with an increase in age*.

## 2. Materials and Methods

### 2.1. Design

This study is based on a cross-sectional descriptive design. This study adheres to the STROBE (Strengthening the Reporting of Observational Studies in Epidemiology) framework to ensure rigorous reporting of observational research [64].

### 2.2. Sample and Setting

With the aim to investigate nationwide drivers of quit intentions among nurses working in various healthcare settings throughout Italy, a maximum variation sampling strategy was used. Registered nurses working in public hospitals, private clinics, specialized care units, community healthcare facilities, and home care services were invited to participate in this research. Such a diverse range of contexts was included with the intention to provide a comprehensive knowledge of the phenomenon.

Participation was voluntary and anonymous. The sample was selected using a convenience-based approach [65]. To avoid bias in convenience sampling, we ensured an equal opportunity for participants to be a part of this research. A survey was distributed at various moments, across settings and regions, to improve diversity [66]. This sampling method has proven to be efficient, cheap, and simple to implement [67,68,69].

Registered nurses were qualified for inclusion in this study if they (a) were employed in either public or private healthcare institutions, (b) collaborated within interdisciplinary healthcare teams, and (c) had at least one year of professional experience. Exclusion criteria applied to nurses (a) working freelance, (b) practising in isolation, (c) serving as float nurses, or (d) returning to work for less than two months after an extended absence.

The required sample size was calculated a priori using Monte Carlo power analyses for mediation models and indirect effects. A target power level of 0.80 was set, with a minimum sample size of 50, a maximum of 1000, and increments of a sample step size of 10 to estimate power. The results indicate that a minimum of 470 participants was sufficient to achieve a statistical significance of *p* < 0.05 (https://schoemanna.shinyapps.io/mc_power_med/, accessed on 10 March 2025). To enhance the robustness of this study, 2000 participants were recruited.

### 2.3. Data Collection

This study was introduced to the Italian Scientific Society for the Direction and Management of Nursing (SIDMI) network. The principal investigators (DI and DT) presented this research to the Directors of Nursing. Those manifesting an interest in this study appointed local representatives to facilitate recruitment among registered nurses at their respective institutions. These representatives actively promoted the survey through multiple outreach efforts with nurse managers and staff, reinforced by regular reminders. Data collection was conducted via an online survey distributed through institutional email addresses. Responses were collected post-COVID-19, between August 2022 and December 2023. The principal investigators and local representatives maintained ongoing communication about participation rates and ensured reminders were issued consistently to enhance engagement.

### 2.4. Measurements

The survey was composed of three sections:

The survey’s initial section collected socio-demographic information, including age, gender, level of education, tenure, employment setting, and the type of healthcare organization.

The second section of the survey assessed registered nurses’ intentions to leave their current unit, healthcare institution, or the nursing profession. This was evaluated using three single-item questions with dichotomous response options: “0” (no, I do not intend to leave) and “1” (yes, I intend to leave within the next six months). Using single-item measures, nurses were then asked to rate their satisfaction with their role, interdisciplinary and peer team, leader, and institution. The answers were rated against a 5-point Likert scale ranging from 0 (very unsatisfied) to 4 (very satisfied), with higher scores indicating greater satisfaction. Cronbach’s alpha for the sample was 0.854, demonstrating satisfactory reliability for the scale.

The third part assessed work environment dynamics. The Questionnaire on Experience and Work Evaluation (QEEW 2.0© SKB) [70] incorporates different instruments included in the model. The scoring for each scale is reported in standardized values ranging from 0 to 100, with higher scores indicating worse measurement significance. For analytical purposes, the researchers utilized converted scores. These scales have been previously validated in the Italian population.

Physical workload was assessed with the Pace and Amount of Work scale, which consists of six items rated on a 4-point Likert scale from 0 (never) to 3 (always), with lower scores indicating a lower physical workload. This unidimensional scale demonstrated good internal consistency with this study’s Cronbach’s alpha of 0.896.

Mental workload was measured using the Mental Workload scale, which includes four items rated on a 4-point Likert scale from 0 (never) to 3 (always), where lower scores indicate less mental strain. This scale is unidimensional. This study’s Cronbach’s alpha is 0.795.

Emotional workload was measured with the Emotional Workload scale, comprising five items rated on a 4-point Likert scale from 0 (never) to 3 (always), where lower scores represent a lower emotional workload. This scale is unidimensional. This study’s Cronbach’s alpha is 0.797.

Autonomy was measured with the Autonomy scale, comprising four items rated on a 4-point Likert scale from 0 (never) to 3 (always), where lower scores represent lower work autonomy. This scale is unidimensional. This study’s Cronbach’s alpha is 0.807.

Inspiration by the leader was measured with the Inspiration by the Leader scale, comprising four items rated on a 4-point Likert scale from 0 (never) to 3 (always), where lower scores represent lower work autonomy. This scale is unidimensional. This study’s Cronbach’s alpha is 0.921.

To evaluate the quality of the relationship with the leader, the LMX-7 scale was used [71]. This scale is unidimensional, consisting of seven items measured using a 5-point Likert scale ranging from 1 (never) to 5 (frequently). The score was calculated as the mean, with higher scores indicating a greater quality of relationships. Cronbach’s alpha for the scale in this study is 0.919. Previous research adapted and translated the scale into Italian [54].

### 2.5. Ethical Considerations

This research received approval from the local Ethics Committee. Following this, the Board of Directors from each participating healthcare organization was contacted to obtain local consent. This study adheres to ethical guidelines, aligning with the principles outlined in the Declaration of Helsinki [72]. Before participating, all individuals were provided with comprehensive details about the research aims, procedures, and data management, and they gave online informed consent. Throughout the study period, participants’ privacy and confidentiality were rigorously maintained. Data access was solely restricted to the research team.

### 2.6. Statistical Analysis

Descriptive statistics were used to summarize demographic and sample data. Continuous variables were assessed for normal distribution and are reported as means with standard deviations (SDs), while ordinal and nominal variables were presented as frequencies and percentages. Non-normal distributed variables are reported as median and quartile ranges. Pearson’s correlation coefficients were used to examine relationships between variables. The initial analysis included checking for linearity, outliers, missing data, homogeneity of variance, and multicollinearity.

To confirm that the items loaded correctly on their respective constructs and that the constructs were distinguishable, a confirmatory factor analysis (CFA) was performed [73]. The CFA was conducted using the unweighted least squares mean and variance (ULSMV) method, treating the intention to leave as a categorical variable. Cronbach’s alpha coefficients were calculated to evaluate the reliability of the scales.

In the next step, we utilized structural equation modelling (SEM) to test the proposed hypotheses. Theoretical considerations guided the selection of independent, dependent, and mediator variables. The model included latent constructs for job satisfaction, quit intentions, and job resources: autonomy, inspiration by the leader, and the leader–member exchange quality. A second-order latent construct was used for job demands. Job demands and job resources (see Figure 1) were specified as the independent variable, job satisfaction as the mediator, and quit intentions as the dependent variable. We performed specific indirect effects tests to examine the mediation effects using the “Model Indirect” procedure in Mplus 8.4.

The goodness-of-fit indices were evaluated following established guidelines from the literature [74,75,76]. For both the CFA and SEM models, we examined several fit indices, including χ^2^, the root-mean-square error of approximation (RMSEA), the standardized root-mean-square residual (SRMR), the comparative fit index (CFI), and the Tucker–Lewis index (TLI). The model fit was considered acceptable based on the following thresholds: RMSEA < 0.08 (acceptable) and <0.05 (good fit), SRMR < 0.05 (good fit), CFI > 0.95 (good fit), and TLI > 0.90 (good fit). Additionally, traditional χ2 statistics were used to assess the model’s overall fit [74,76]. Standardized regression coefficients (β) and the coefficient of determination (R^2^) are also reported.

All statistical tests were two-tailed, with a significance level set at *p* < 0.05. Missing data for each variable were examined and handled using available case analyses. Data analyses were conducted with the SPSS software version 27.0 (IBM Corp., 2019, Armonk, NY, USA) and MPLUS version 8.4.

## 3. Results

### 3.1. Participants

Of the 2000 participants invited to complete the survey, 1745 responded, yielding a response rate of 87.2%. The missing data were minimal, accounting for less than 1%. Table 1 outlines the socio-demographic characteristics of the participants. The majority of the sample were female (76.6%), with a mean age of 43.7 years (range: 22–66) and a mean tenure of 18 years (SD = 11.3; range 1–45). Most responses came from nurses working primarily in public healthcare settings (73.8%) and mainly in hospitals (72.8%). A significant proportion (61.2%) worked rotating day/night shifts, and 73.2% held a bachelor’s degree in nursing as their highest educational qualification. These data align with the characteristics of nurses in Italy. Indeed, the country is experiencing a significant shortage of nurses due to the high number of nurses reaching retirement age. Like other nations, Italy’s nursing profession is predominantly composed of females.

The description of the study variables is presented in Table 2. The sample reported mean level scores for job satisfaction (mean, 2.6; SD, 0.8); physical workloads (mean, 43.6; SD, 21.7); emotional workloads (mean, 51.5; SD, 19.1); and the quality of leader–member exchanges (mean, 3.5; SD, 0.8). While the workload scores are considered acceptable, the moderate levels of job satisfaction and the quality of leader–member relationships highlight the challenging nature of the work environment. High workload levels are evidenced for the mental load (mean, 88.8; SD, 15.1), indicating significant cognitive demands. Higher level scores were evidenced for nurses’ autonomy (mean, 27.3; SD, 7.4) and feeling inspired by the leader (mean, 39.5; SD, 28.5), suggesting positive levels of job resources that may buffer some of the adverse effects of workload.

However, the findings on quit intentions are concerning, with a notable proportion of nurses considering leaving their unit (26.5%), institution (22.2%), or even the profession altogether (17.7%), highlighting the urgency of addressing these challenges to improve retention.

### 3.2. Correlations

The analysis identified significant yet small correlations between participant characteristics and study variables (Table 3). Job satisfaction exhibited positive correlations with higher education levels and day shifts, indicating greater satisfaction in these groups. Quit intentions were negatively associated with age and day shifts, suggesting lower turnover intentions among older nurses and those working day shifts. However, higher education levels were linked to increased turnover intentions. Job demands were inversely related to age and day shifts, reflecting lower perceived workloads, while higher demands were reported in private institutions. Similarly, job resources improved with age and day shifts but were reduced in private healthcare settings. All significant Pearson correlation coefficients were low (refer to Table 3 for details).

Nurses’ intentions to quit were negatively correlated with job resources and positively correlated with job demands, with stronger associations observed for job satisfaction (r = −0.49; *p* < 0.01) and autonomy (r = −0.60; *p* < 0.01). This finding indicates that enhancing these factors could significantly contribute to improving retention. Job satisfaction, in turn, showed positive correlations with job resources and negative correlations with job demands, with higher coefficients observed in job resources. This aligns with previous research and reinforces the principles of the JD-R framework. Additionally, job demands were inversely correlated with job resources, though the relationships displayed small coefficients. Detailed statistical values are presented in Table 4.

### 3.3. Measurement Model

The CFA was conducted using the ULSMV estimation method, confirming that the specified variables were distinctly and consistently loaded onto the six identified factors. The loadings for job satisfaction ranged from 0.65 to 0.85, while the intention to leave had loadings ranging from 0.64 to 0.86. Job demands were specified as a second-order factor, with the three dimensions (physical, mental, and emotional workloads) having loadings ranging from 0.40 to 0.70. Job resources (autonomy, inspiration by the leader, and the quality of leader–member exchanges) were tested for a second-order factor model, but the results are not significant. See Appendix A for additional information.

### 3.4. Mediation Model

The SEM conducted to examine the relationship between nurses’ job demands, job resources and job satisfaction and nurses’ quit intentions presented satisfactory fit indices: χ2 (49; N = 1745) = 1849.36; *p* < 0.001; CFI = 0.95; TLI = 0.93; and RMSEA = 0.04 (90% CI = 0.03–0.04). Figure 2 presents all the effects observed.

Nurses’ job demands negatively influenced nurses’ job satisfaction (β = −0.24; *p* < 0.001) and were positively related to nurses’ quit intentions (β = 0.20; *p* < 0.001). This result supports Hypotheses 1 and 3, and it is consistent with the existing nursing literature. Nurses’ job resources, in turn, positively influenced nurses’ job satisfaction (autonomy β = 0.15; *p* < 0.001; inspiration by the leader β = 0.35; *p* < 0.001; and quality of relationship leader–member exchange β = 0.19; *p* < 0.001) and reduced nurses’ quit intentions (respectively, β = 0.12, β = 0.32, and β = 0.15; *p* < 0.001). This finding confirms Hypotheses 2 and 4 and aligns with the existing body of literature. Nurses’ job satisfaction presented inverse effects with nurses’ quit intentions (β = −0.42; *p* < 0.001), confirming Hypothesis 5. Furthermore, inverse effects were observed between quitting intentions and age: older nurses were associated with reduced turnover intentions (β = −0.11; *p* < 0.001). This finding supports Hypothesis 7 and aligns with the existing literature, which highlights that younger nurses are at a higher risk of turnover, emphasizing the need for targeted attention from nurse managers.

Hypothesis 6 proposed the mediation analysis. Nurses’ job satisfaction did not mediate the relationship between job demands and job resources and nurses’ quit intentions (total effect β = −0.42; *p* < 0.001; indirect effect β = 0.00; *p* < 0.001; and direct effect β = −0.42; *p* = 0.001). This finding aligns with the literature supporting the inclusion of job satisfaction as a predictor within the JD-R framework. Indeed, in this study, the relationship between job satisfaction and the intention to leave exhibited the highest coefficient in the model, supporting this suggestion in the previous literature.

Based on the coefficients presented in Figure 2, we can observe that aside from job satisfaction, inspiration from the leader emerges as the strongest predictor of nurses’ turnover intentions and job satisfaction in this sample. This underscores the critical role of nurse managers in retaining nursing staff and highlights the importance of applying transformational leadership styles in workplace settings to enhance retention.

Overall, the model explained 47% of the variance in the intention to leave results. Further details are presented in Table 5.

## 4. Discussion

In an effort to describe the determinants of job turnover intentions among Italian nurses, this study explored individual and interpersonal interactions and job-related factors influencing quitting intentions and examined the mediating role of job satisfaction in these associations. Based on the JD-R theory, the variables were grouped into job demands and job resources.

### 4.1. Job Demands

This study examined job demands, focusing on physical, mental, and emotional workloads. The nurses participating in this study reported moderate levels of physical and emotional workloads and high mental load levels. These findings are consistent with previous results from another study conducted by the research team in Italy [22]. Although the workload levels were not critical, significant effects of job demands on job satisfaction and the intention to leave were observed. High job demands in nursing were associated with reduced job satisfaction and increased nurse turnover intentions. These results align with findings from studies conducted prior to the COVID-19 pandemic [6]. Therefore, this study supports the claim that a high workload continues to be a key determinant of job satisfaction and nurses’ decisions to stay or leave their positions. This factor remained consistently influential even after the COVID-19 pandemic, highlighting its persistent role as a primary driver of turnover intentions in the nursing profession.

According to the JD-R model, job demands drive the health-impairment process [16]. This is further supported by the nursing workload literature, which highlights that high physical workloads can lead to fatigue, stress, exhaustion, and even disabilities—all well-documented predictors of turnover intentions [77,78]. Furthermore, mental and emotional demands, such as the stress of patient care and emotional labour, exacerbate the risk of burden, job dissatisfaction, and exhaustion, contributing to reduced performance and increased turnover intentions [11,32,79,80,81,82,83]. These findings from international research applying the JD-R and SET frameworks align with the results of our study. Indeed, Italian nurses, like nurses from countries such as the United States, China, and New Zealand, have reported similar trends, where high physical, mental, and emotional workloads correlate with increased job dissatisfaction and turnover intentions [13,20,80]. Despite differences in healthcare systems and staffing models, the persistent influence of job demands on nurse retention underscores its universal relevance across diverse contexts.

Given the substantial evidence on the impact of workloads on nursing outcomes, it is crucial for institutions to systematically assess and monitor this phenomenon. Italy is currently experiencing a significant shortage of nurses. Increasing workloads places additional strain on the existing nursing workforce and heightens the risk of turnover [84]. Consequently, monitoring workloads and mitigating job demands is paramount. Addressing these factors is essential to promote a sustainable nursing workforce and ultimately to improve patient care outcomes in Italy and beyond.

### 4.2. Job Resources

This study examined key job resources based on the JD-R theory, focusing on nurses’ autonomy, inspiration by the leader, and the quality of relationship with the leader (leader–member exchanges). The findings underscore that higher levels of job resources significantly boost job satisfaction and reduce turnover intentions, confirming the motivational process of job resources as outlined in the JD-R and SET framework [16]. Among the job resources examined, inspiration by the leader and the leader–member relationship quality had the strongest effects. This result reinforces existing evidence on the critical role of job resources in nurse retention [2,32,53], while advancing the field by shedding light on the influence of inspiration by the leadership—a variable that has been underexplored in the literature. By demonstrating its significance among Italian nurses, these findings broaden the understanding of leadership’s impact on workforce retention, underscoring the importance of adopting supportive and motivational leadership strategies in healthcare settings. Indeed, factors such as supportive leadership, autonomy, meaningful work, positive workplace relationships, and opportunities for personal development are well-documented in the literature as protective against burnout and turnover while fostering job satisfaction [3,4,9,27,60,61]. When leaders adopt a transformational leadership style, they motivate and inspire nurses; lead by example; and foster teamwork, optimism, and overall employee well-being [32,85]. This approach positively influences nurse satisfaction and retention [2]. This study’s insights, therefore, not only contribute to the present academic discourse but also offer practical implications for fostering a more engaged and stable nursing workforce.

A healthy nursing workforce requires healthcare managers to prioritize the well-being of their staff, sharing responsibility for building a sustainable workforce [49]. Nurse managers, in particular, must cultivate advanced competencies in leadership, human resource development, and communication, while fostering trust and high-quality relationships within their teams to reduce turnover intentions and promote a flourishing workplace [28,29,37,38,39,40,62]. Research across different contexts, regardless of profession, national setting, or healthcare system, confirms that strong interpersonal connections and a supportive work environment remain essential drivers of employee retention [32,53]. Moreover, in the post-pandemic era, these aspects have become even more critical, as the crisis has underscored the importance of emotional and social support in sustaining workforce engagement and well-being [2]. However, research on leadership styles and their impact on nursing, patient, and organizational outcomes remains limited in Italy. The findings from this study, particularly regarding the explored job resources, are among the first to examine such variables in Italian nurses. Consequently, these results can inform targeted educational and organizational interventions to enhance the nursing workforce and strengthen nurse manager competencies.

### 4.3. Job Satisfaction

Nurses’ job satisfaction was not confirmed as a mediator in the tested model but emerged as the most significant determinant of turnover intentions among Italian nurses. This finding contrasts with much of the organizational research, where job satisfaction is often identified as a mediating variable in models predicting turnover intentions [63]. For example, previous studies have shown that job satisfaction mediates the relationship between decent work and turnover intentions [27], perceived workloads, work–life balance, and the intention to leave [63], as well as the meaning of work and turnover intentions, although not specifically for workloads [64]. Furthermore, job satisfaction, work engagement, and burnout have been found to mediate the relationships between job resources, job demands, and turnover intentions among healthcare professionals in hospitals [58]. Similarly, job satisfaction mediates the link between leader–member exchanges and turnover intentions [59] or between job stress, presenteeism, and turnover intentions [60].

However, this result is not unprecedented, as other research has similarly observed the direct effect of job satisfaction on turnover intentions without a mediating role [9,65]. The absence of mediation may be due to the strength of the direct effect of job satisfaction on turnover intentions. Low job satisfaction is strongly associated with increased turnover intentions. For example, recent research conducted among healthcare professionals in northern Italy tested the mediation effect of job satisfaction between relational coordination and willingness to stay. The authors found that job satisfaction was not a mediator in this relationship. Instead, it was confirmed as a significant predictor of the intention to stay, with a strong effect size [61]. This supports the rationale for continuing to use extended models that include job satisfaction as a predictor when assessing employees’ willingness to remain in their roles.

Job satisfaction is a direct expression of a nurse’s affective and cognitive appraisal of their work experience [4,63,65]. It encompasses both emotional well-being and the intrinsic rewards derived from meaningful work and opportunities for personal development [9]. Job satisfaction, burnout, and turnover intentions are the outcomes of intricate interactions between job demands and job resources [12]. The inverse relationship between job satisfaction and turnover intentions is well-documented [9,62]. Despite its predictive reliability, job satisfaction is only one facet of workplace well-being [9]. Limiting the analysis to job satisfaction overlooks broader elements, such as work meaningfulness or professional fulfilment, which may also significantly influence turnover intentions [9]. Future research should further explore whether job satisfaction can serve as a proxy for broader aspects of well-being or whether alternative constructs might better explain nurses’ decisions to leave their positions.

### 4.4. Individual Factors

A recent literature review spanning the past decade on the determinants of turnover intentions underscores that individual factors, particularly the role of generational differences in shaping turnover behaviour, remain insufficiently examined [86]. This study addresses this critical gap in the literature by offering a more nuanced exploration of the impact of individual-level factors on turnover intentions, thereby advancing the theoretical understanding and providing empirically grounded insights for both the academic discourse and organizational practice. Indeed, the results of our study align with and extend the existing literature on factors influencing nurses’ job satisfaction and turnover intentions. Consistent with prior research [54], we found that younger nurses were more likely to have higher turnover intentions than older nurses. In previous studies [2,54], younger nurses reported more job stress than experienced nurses due to workloads, difficulty in interpersonal connections, and a lack of work skills when adjusting to a new setting and learning new tasks. This limited development of relationship-building and communication skills, which is particularly pronounced among Generation Z—often referred to as “digital natives”—can contribute to heightened levels of stress and dissatisfaction within a profession that fundamentally depends on effective teamwork and patient-centred interactions [87]. Furthermore, Generation Z nurses exhibit distinct workplace expectations that diverge from those of preceding generations. They place a strong emphasis on work–life balance, mental well-being, and meaningful engagement in their professional roles [88,89,90]. When these expectations remain unmet, they demonstrate a greater propensity to consider turnover, highlighting the need for organizational strategies that align with their unique values and priorities.

Nursing research has shown that younger nurses, particularly those under 39 years old, are more likely to consider leaving both their hospital and the healthcare profession [2,62]. This finding is consistent with previous studies conducted among Italian hospital nurses [6]. However, it is important to note that age-related turnover is not exclusive to younger nurses. During the COVID-19 pandemic, older workers also demonstrated higher intentions to leave, suggesting that all workers can be potential quitters, depending on contextual factors [2]. Age-related effects in the workplace are multifaceted and may include complex perceptual, interpersonal, and intergroup dynamics. For instance, despite regulations aimed at preventing age discrimination, older workers often face challenges stemming from age-related biases [55]. These biases can lead to feelings of undervaluation and exclusion, further exacerbating their intention to leave the profession.

The findings of this study highlight the challenges faced by nurses in early career stages, particularly those from Generation Z, and underscore the importance of tailoring management strategies to address specific age-related and tenure-related needs [2,4,55]. For instance, healthcare organizations could introduce structured mentorship programs designed to facilitate intergenerational collaboration and knowledge transfer between younger and more experienced nurses, thereby enhancing mutual understanding and skill sharing [87]. Furthermore, offering tailored support for digital natives, such as specialized training in interpersonal communication and stress management techniques, could effectively reduce turnover intentions among younger nursing staff [87]. These initiatives would not only address the unique challenges faced by Generation Z nurses but also promote a more cohesive and resilient workforce.

Beyond age, other individual factors may also influence turnover intentions. Findings from a recent systematic review [2] indicate that nurses’ turnover intentions increase with factors such as work tenure, extended work schedules, low wages, understaffing, direct care responsibilities, and rotating shifts. Additionally, a higher risk was identified among females and those with family responsibilities [5]. In this study, rotating shift schedules were found to be significantly associated with an increase in nurses’ intentions to leave. This finding aligns with prior research indicating that frequent night shifts are positively correlated with turnover intentions, emphasizing the necessity of allocating night shifts strategically to mitigate their impact on nurses’ well-being and retention [4,61,66]. One’s educational level also emerged as a significant factor, with higher-educated nurses being more likely to consider leaving, identifying education as a push factor for turnover. This points to a potential disconnect between advanced qualifications and career opportunities or workplace satisfaction [4,66]. Additionally, gender and flexible scheduling were found to significantly impact retention [4,66], which is not confirmed in this research. These findings collectively reinforce the multifaceted nature of turnover intentions in nursing, highlighting the need for holistic and targeted intervention strategies by nurse administrators to effectively address job demands and resources.

Moreover, according to what the JD-R and SET frameworks describe, workloads, particularly mental and physical demands, continue to play a pivotal role in shaping turnover intentions, becoming “stressors”, with private institutions and rotational shifts exacerbating these effects. Our findings on job resources, such as autonomy, leader inspiration, and the leader–member exchange quality, align with the JD-R and SET models and existing evidence emphasizing transformational leadership’s protective and supportive role and meaningful work environments [53,65]. These factors, such as satisfaction and retention, were shown to increase positive actions described in SET, further underscoring the need for managerial efforts to foster autonomy and strong, trust-based relationships within teams. Addressing these predictors systematically—while paying special attention to the nuances introduced by nurses’ age, experience, and educational background—offers a promising pathway to improving retention and sustainability in the nursing workforce.

The healthcare sector has undergone significant changes in recent years, with the COVID-19 pandemic exacerbating workforce challenges and fuelling a “Great Resignation” as employees prioritize work–life balance [4,15]. In recent years, interest in this issue has grown significantly, underscoring its rising importance in healthcare workforce research within the country. Studies indicate that 77% of healthcare resignations were considered preventable and attributed to factors within the employer’s control [91]. Nursing retention is particularly affected by workplace conditions, with high workloads posing substantial risks [92] and strong job resources improving outcomes [93].

### 4.5. Implications for Managerial Practice

Leadership plays a crucial role in cultivating supportive and healthy work environments, which are essential for minimizing staff turnover and improving organizational performance [81]. Effective leadership fosters trust, commitment, and job satisfaction, creating a foundation for workforce stability and operational success.

Nurse managers, in particular, must focus on promoting the overall thriving of their teams, encompassing mental, physical, and social well-being [93]. Interventions need to be targeted and focused on expressed and observed needs [58].

Considering the important shortage of nurses in Italy and the findings from this study, to support nurses in addressing job demands and workloads, therefore, to reduce fatigue, strain, and stress, nurse managers in Italian healthcare institutions need to ensure safe patient care through adequate nurse-to-patient ratios, safe patient-handling equipment, or flexible scheduling to improve nurses’ work–life balance and enhance their retention [71]. Additionally, to empower nurses’ job resources, nurse managers can adopt different strategies. For example, facilitating continuous training improves nurses’ abilities to master their work, develop competencies, enhance self-efficacy, and strengthen decision-making skills, ultimately promoting greater autonomy [61]. Similarly, enhancing mentorship and coaching supports nurses in building confidence, improving patient care, and contributing meaningfully to their workplace [61]. Furthermore, nurse managers should adopt clear communication to create healthy workplaces, discuss errors as learning opportunities, and recognise and reward nurses’ professional and personal achievements [93]. This will reinforce nurses’ engagement and satisfaction and increase retention. Furthermore, one of the most valuable leadership behaviours nurse managers can adopt is providing personal attention to nurses. This includes activities such as showing interest in their personal issues or concerns, goals, professional growth, and career aspirations, as well as caring for the team, being present and accessible, and fostering team building [80]. These behaviours help instil trust, support, and a sense of being valued among nurses. All these leadership behaviours are inherent to the transformational leadership style.

To adopt these behaviours, nurse managers need to be competent in their leadership. In Italy, limited research exists on nurse managers’ preparedness and the impact of their competencies on workforce well-being [68,70]. Therefore, this research contributes to advancing knowledge in the field. We recommend nurse managers in Italy to actively engage with their nurses through interventions focused on personal resource development, leadership training, health promotion, job crafting, and mindfulness practices [2].

The early identification of potential quitters is critical to ensure sustainability amidst the ongoing nursing shortage [15]. Effective strategies must focus on reducing turnover, enhancing the profession’s appeal, and retaining young nurses, which are essential for the resilience of healthcare organizations [11,71].

### 4.6. Theoretical Implications

The results of this study align with the JD-R theory, reinforcing the notion that job resources can mitigate job demands and lead to improved outcomes for nurses. Furthermore, this study’s findings support the theoretical assertion regarding the negative impact of job demands on nurse outcomes, as evidenced within the nursing population across Italy.

The findings of this study suggest a potential extension of the JD-R theory, emphasizing the role of individual factors—such as age, tenure, education, work schedules, and organizational context—in shaping job demands and resources. These variables should be integrated into both the JD-R and SET frameworks, particularly in the context of workforce retention challenges among Generation Z and future labour markets. Given the growing emphasis on workplace well-being, mental health, and work–life balance, these factors are crucial in ensuring nurse retention and overall job satisfaction.

This perspective offers a significant contribution to the research on job resources, advocating for a paradigm shift in how job design and work engagement strategies are developed. Institutions must adopt a holistic approach, combining structural changes with strategies that foster a resource-rich work environment and an optimal employee–organization fit. Theoretical models examining these dynamics should further explore how to enhance employees’ autonomy and control over their roles, reinforcing their sense of purpose and engagement in the workplace.

Workload in nursing practice has been extensively studied and debated in the literature due to their well-documented negative effects on nurses’ outcomes. Nursing research has focused on strategies to control and reduce workloads for decades. However, when interpreting workload as a job demand within the JD-R framework, it becomes evident that other variables may hold greater relevance for nursing practice and outcomes. Our study model observed the strongest effect in the job resource “inspiration by the leader”. This finding highlights the need to prioritize the exploration of job resources and their impact on nurses in research and clinical practice, suggesting a shift in focus from solely mitigating demands to enhancing supportive resources.

Overall, in relation to the JD-R framework, this study confirms the relationships and directions of effects as outlined in the theory and supported by the previous nursing literature. Notably, inspiration by the leader emerges as a job resource warranting further exploration in future research, as it has received comparatively less attention.

The “inspiration by the leader” variable can also be analysed through the lens of Social Exchange Theory. From this viewpoint, it may be categorized as an inactive exchange. Unlike deliberate and strategic leader actions, inspiration by the leader is conveyed through involuntary psychological exchanges in the leader–nurse relationship. Proximity to a leader who naturally adopts positive relational behaviours fosters inactive/involuntary positive exchanges, which, in turn, significantly impact nurses’ experiences and outcomes.

Healthcare systems and societies experienced profound changes during and after the COVID-19 pandemic. These shifts necessitated a renewed application and adaptation of the JD-R and SET frameworks to address emerging challenges. The findings of this study contribute valuable insights that may inform and advance the ongoing development of these theoretical frameworks.

The findings of this study highlight the critical importance of job resources in shaping workplace relationships for nurses. Leaders must develop new skills and adapt to the evolving dynamics of modern workplace interactions and relationships. Relationships have become central to teamwork and workplace well-being in the post-COVID era, profoundly influencing modern exchange processes.

### 4.7. Strengths and Limitations

This research possesses several noteworthy strengths. It provides empirical evidence on critical factors influencing nurses’ decisions to leave the profession, focusing on variables that have been underexplored in previous studies [71,72]. The extensive sample size, diverse geographical distribution, institutional variability, and work setting heterogeneity comprehensively represent the national nursing landscape in Italy. Moreover, the high response rate and low frequency of missing data underscore the strong engagement of nurses with the topic.

Compared to previous studies, this research is the first to examine the explored variables within the JD-R and SET frameworks in the Italian context, specifically focusing on nurses’ intentions to leave the profession. Furthermore, the existing literature presents inconsistent findings regarding the mediating role of job satisfaction in the relationship between job demands/resources and turnover intentions. In the Italian sample of nurses, no significant mediation effect was observed, a notable and unexpected result that may indicate the need for further investigation. This finding highlights a critical gap in understanding and suggests the need for additional research in the field of organizational well-being. Future studies should delve into the underlying reasons for this phenomenon and identify other potential factors that may hold greater predictive power in explaining nurse turnover and workforce attrition. Such insights could refine theoretical models and inform more effective retention strategies tailored to the unique dynamics of the nursing profession.

This study has several limitations that warrant consideration. First, the use of a cross-sectional design limits our ability to establish causal relationships between the variables studied. While the findings provide valuable insights into the associations explored, future longitudinal research and Cox regression analysis is needed to confirm causal pathways. For example, future research incorporating both cross-sectional and longitudinal data could employ Poisson regression with robust variance and log-binomial regression to enhance the accuracy of estimates. These methods offer a more suitable alternative to logistic regression for analysing binary outcomes in cross-sectional studies, as the prevalence ratio provides a more intuitive and interpretable measure, particularly for non-specialist audiences. However, careful consideration is necessary to mitigate potential estimation challenges that may arise in specific cases.

Second, the reliance on self-report measures introduces potential biases, such as social desirability effects and recall bias, which may influence the accuracy and objectivity of the responses. While subjective measures are valuable for capturing individual perceptions, the inclusion of longitudinal data or multiple data sources, such as actual turnover rates, would enhance this study’s internal and external validity. Efforts were made to mitigate this issue through anonymous survey administration; however, the inherent subjectivity of self-reports remains a limitation.

Third, the convenience sampling approach used in this study may have introduced a sampling bias. An over-representation of certain institutions or settings could limit the generalizability of the findings to other contexts or nurses in Italy. Although the sample was diverse in terms of demographic and professional experience and geographic areas, it may not fully represent the broader nursing workforce. Future research should aim for more representative sampling to ensure the robustness of conclusions.

Additionally, this study examined only a subset of variables associated with nurse turnover, potentially overlooking other relevant factors. For instance, distinctions between turnover rates among nurses in permanent versus temporary positions were not explored. Similarly, differences in turnover intentions between nurses in public and private healthcare sectors, or specific intentions related to unit, organizational, or professional turnover, were not comprehensively analysed. This study did not test sensitivity analyses, which could have provided a more comprehensive understanding of the examined relationships. This choice was made to maintain a focused scope and align with the primary research objectives. However, incorporating alternative models and sensitivity analyses in future studies could enhance the reliability and applicability of the findings.

Despite these limitations, this study contributes valuable evidence to understanding the determinants of turnover intentions among nurses and offers a foundation for further research in this area.

### 4.8. Future Research

Future research should aim to validate the predictive capability of the proposed model for actual occupational turnover. Longitudinal studies are particularly needed to establish causality and enhance our understanding of how specific variables influence nurses’ quit decisions over time (e.g., safe environment, control over the work environment, etc.). Investigating leadership styles and other underexplored factors could provide deeper insights into turnover intentions’ dynamics. Examining the interplay between organizational culture, team cohesion, and work–life balance may uncover critical leverage points for intervention fitting the nursing labour force.

Given the pressing challenges the nursing workforce faces, future research must prioritize exploring factors that enhance the profession’s attractiveness and foster retention. Such efforts are pivotal for designing strategies that enable healthcare organizations to sustain skilled and committed nursing staffing, ultimately improving patient care quality and organizational stability. Future research should explore additional, less-studied factors influencing nurses’ turnover intentions to improve retention, organizational sustainability, and patient care quality.

## 5. Conclusions

Our study provides valuable insights into the factors driving nurses’ decisions to leave the profession. We found that high physical, mental, and emotional job demands are linked to increased dissatisfaction and turnover intentions. In contrast, job resources like decision-making autonomy, inspiration by the leader, and positive leader–member exchanges enhance job satisfaction and retention. Job satisfaction, however, did not mediate this relationship.

These findings highlight that the factors influencing Italian nurses’ intentions to leave or stay are closely tied to workplace characteristics and the quality of nursing leadership. Therefore, nurse managers must adopt transformational leadership practices to foster healthy work environments, strengthen teams, and ensure nurses feel valued and motivated, ultimately enhancing job satisfaction and retention. By building strong relationships and providing personal attention to their staff, nurse managers can empower and retain nurses, particularly new recruits, ensuring adequate staffing levels and maintaining high-quality care across the country. Future research should explore additional, less-studied factors influencing nurses’ turnover intentions to improve retention, organizational sustainability, and patient care quality.

## Figures and Tables

**Figure 1 healthcare-13-00692-f001:**
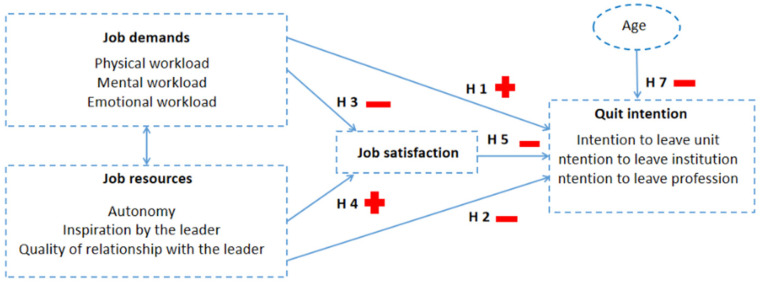
The hypothesis-based research model.

**Figure 2 healthcare-13-00692-f002:**
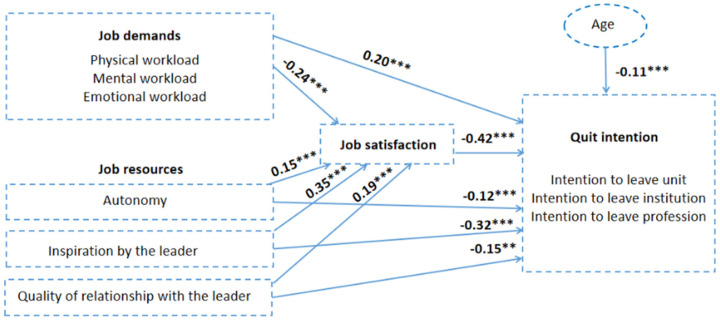
Results of the tested model: the mediation of job satisfaction. *Note*: the coefficients refer to standardized effects. ** *p* < 0.01; *** *p* < 0.001.

**Table 1 healthcare-13-00692-t001:** Sample characteristics (N = 1745).

Variables	N (%)
**Service**	
Public	1288 (73.8)
Private	457 (26.2)
**Age** (mean; SD)	43.7 (11.7)
**Sex**	
Female	1137 (76.6)
Male	408 (23.4)
**Work experience**	
Years (mean; SD)	18.3 (11.3)
**Educational background**	
BSc or equivalent title	1277 (73.2)
Postgraduate certificate after BSc	330 (18.9)
Master of Science	133 (7.6)
Postgraduate certificate after MSc	5 (0.3)
**Shift**	
Rotation shifts (day/night)	1068 (61.2)
Day shift only	677 (38.8)

Legend: SD = standard deviation; BSc = Bachelor of Science in nursing; MSc = Master of Science in nursing.

**Table 2 healthcare-13-00692-t002:** Study variables’ description (N = 1745).

Variables	N (%)
**Intention to leave the unit**	
yes	462 (26.5)
**Intention to leave the institution**	
yes	387 (22.2)
**Intention to leave the profession**	
yes	304 (17.7)
**Job satisfaction** (mean; SD)	2.6 (0.8)
**Physical workload** (mean; SD)	43.6 (21.7)
**Mental workload** (mean; SD)	88.8 (15.1)
**Emotional workload** (mean, SD)	51.5 (19.1)
**Autonomy** (mean; SD)	27.3 (7.4)
**Inspiration by the leader** (mean; SD)	39.5 (28.5)
**Leader member exchange** (mean; SD)	3.5 (0.8)

Legend: SD = standard deviation.

**Table 3 healthcare-13-00692-t003:** Correlations of study variables with socio-demographic information (N = 1745).

	Satisfaction	Quit Intention	Autonomy	Inspiration by Leader	Quality LMX	Physical Workload	Mental Workload	Emotional Workload
Age	−0.04	−0.09 **	0.06 **	0.07 **	0.09 **	−0.13 **	−0.13 **	−0.08 **
Sex	−0.01	0.01	0.01	0.01	0.04	−0.03	−0.03	−0.04
Education	0.08 *	0.08 **	0.01	0.04	−0.06 *	−0.02	−0.04	−0.02
Tenure	0.03	−0.09 **	0.07 **	0.05 *	−0.01	−0.12 **	−0.12 **	−0.06 *
Shift work	0.07 **	−0.09 **	0.03	0.13 **	0.11 **	−0.10 **	−0.10 **	−0.13 **
Service	0.01	0.01	−0.02	−0.07 **	−0.08 **	0.10 **	0.11 **	0.03

*Note*: the numbers refer to Pearson’s correlation coefficient; two-tailed; * *p* < 0.05; ** *p* < 0.01.

**Table 4 healthcare-13-00692-t004:** Correlations between the study model variables (N = 1745).

	2	3	4	5	6	7	8
1 Quit intention	−0.49 **	0.34 **	−0.03	0.18 **	−0.60 **	−0.36 **	−0.29 **
2 Satisfaction	-	−0.37 **	0.04	−0.24 **	0.40 **	0.56 **	0.53 **
3 Physical workload	-	-	0.20 **	0.43 **	−0.02	−0.26 **	−0.23 **
4 Mental workload	-	-	-	0.26 **	−0.01	−0.06 **	−0.05
5 Emotional workload	-	-	-	-	−0.02	−0.18 **	−0.13 **
6 Autonomy	-	-	-	-	-	0.13 **	0.17 **
7 Inspiration by leader	-	-	-	-	-	-	0.82 **
8 Quality LMX	-	-	-	-	-	-	-

*Note*: the numbers refer to Pearson’s correlation coefficient; two-tailed; ** *p* < 0.01.

**Table 5 healthcare-13-00692-t005:** Path estimates and indirect effects of the mediation model.

	*β*	*p*	SE	95% CI
**Direct Effect**				
Job Demands→Job satisfaction	−0.24	<0.001	0.002	−0.020; −0.014
Job Demands→Quit Intentions	0.20	<0.001	0.003	0.009; 0.021
Autonomy→Job satisfaction	0.15	<0.001	0.001	0.004; 0.007
Inspiration by leader→Job satisfaction	0.35	<0.001	0.001	0.008; 0.012
Quality LMX→Job satisfaction	0.19	<0.001	0.029	0.131; 0.244
Autonomy→Quit Intentions	−0.12	<0.001	0.001	−0.007; −0.002
Inspiration by leader→Quit Intentions	−0.32	<0.001	0.002	−0.014; −0.006
Quality LMX→Quit Intentions	−0.15	<0.01	0.052	−0.261; −0.057
Job satisfaction→Quit Intentions	−0.42	<0.001	0.056	−0.563; −0.343
Age→Quit Intentions	−0.11	<0.001	0.003	−0.014; −0.004
**Indirect Effect**				
Job satisfaction→Quit Intentions	0.00	0.000	0.000	0.000; 0.000
**Total effect**				
Job satisfaction→Quit Intentions	−0.42	<0.001	0.056	−0.563; −0.343

Note: *p*, level of significance; SE, standard error; CI, confidence interval. Additional details can be found in Appendix A.

## Data Availability

The datasets presented in this article are not readily available, because the data are a part of an ongoing study. Requests to access the datasets should be directed to the corresponding authors.

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
