# Peer review of "Exploring Nurses’ Quit Intentions: A Structural Equation Modelling and Mediation Analysis Based on the JD-R and Social Exchange Theories"

_healthcare, 2025, doi:10.3390/healthcare13070692_

Round 1
Reviewer 1 Report
Comments and Suggestions for Authors
- Abstract: please add more details regarding the methodology, such as sample characteristics and statistical methods.
- Introduction: Please include more up-to-date references to enhance the rationale for using JD-R and social exchange theories.
- Methods: Explain potential biases from the convenience sampling approach, such as overrepresentation from certain institutions; consider adding a justification for not testing alternative models or conducting sensitivity analyses.
- Results: please provide additional interpretations of Figure 2, emphasizing the implications of nonsignificant mediation effects.
- Discussion: Please add sub-headings for better readability. Address why job satisfaction did not mediate the relationships as hypothesized. Consider Expanding on practical recommendations, particularly how healthcare managers can implement interventions targeting job resources.
- Pay attention to some typographical errors before submitting your revision.
Reviewer 2 Report
Comments and Suggestions for Authors
Dear authors
Thank you for having me in reviewing your manuscript. Overall, my evaluation of this manuscript is as follows.
Limited novelty: The study primarily validates known relationships without contributing new theoretical insights.
Descriptive findings: A lack of critical analysis reduces the study's impact
Underdeveloped theoretical engagement: The JD-R and SET frameworks are not leveraged to their full potential.
Here are the detailed comments:
Title, Abstract, and Keywords
1. The title omits mention of the theoretical frameworks (e.g., JD-R and SET), which could enhance its specificity.
2. The abstract lacks specific details about the unexpected findings, such as the lack of mediation by job satisfaction.
3. The keywords do not include "Social Exchange Theory" or "Job Demands-Resources," which are central to the study.
Introduction
- Lack of a clear main debate: While the study's relevance is established, the introduction does not present a focused debate or gap in the literature.
- Implicit research questions: The research questions are embedded within the discussion of aims and hypotheses but are not explicitly stated.
- Limited novelty articulation: The introduction does not clarify how this study uniquely contributes to the body of knowledge.
Literature Review
- Lack of critical synthesis: The review reads as a compilation of studies without synthesizing themes or identifying contradictions.
- Missing engagement with international comparisons: How these frameworks and factors operate differently in Italy versus other countries is not explored.
- Inconsistent thematic transitions: The discussion shifts abruptly between individual factors, job demands, and job resources.
Methodology
Shortcomings:
- Over-reliance on cross-sectional design: Limits causal inferences.
- Potential for self-report bias: The reliance on subjective measures may compromise data accuracy.
- Insufficient detail on variable selection: While variables are linked to JD-R and SET, the rationale for including or excluding specific dimensions is not fully explained.
Results
- Descriptive focus: The findings are heavily descriptive and lack interpretative depth.
- Limited exploration of unexpected results: For instance, the lack of mediation by job satisfaction is mentioned but not critically examined.
- Overloaded tables: Some tables include too much data, making them difficult to interpret.
Discussion
- Superficial analysis: Key findings, such as the age effect on quit intentions, are mentioned without delving into underlying mechanisms.
- Lack of theoretical implications: The discussion does not critically engage with how findings extend or challenge JD-R and SET.
- Overemphasis on practical implications: Theoretical contributions are underdeveloped, and recommendations for practice are too generic.
Conclusion
- Lacks future directions: It does not provide a clear roadmap for future research.
- Overgeneralized recommendations: Fails to tailor suggestions to the Italian healthcare context.
This manuscript has great potential to contribute meaningfully to the literature on nurse turnover intentions. Addressing these areas will not only strengthen the theoretical and practical contributions but also improve the manuscript’s overall coherence and impact.
Thank you for your hard work and contribution to this critical field. I look forward to seeing a revised version of your manuscript.
Regards.
Regarding the quality of English, here are my comments:
- Language Polishing:
- Engage a professional editor or use advanced grammar tools to correct errors and refine phrasing.
- Streamline Sentences:
- Aim for shorter, more direct sentences to improve readability.
- Synonyms and Variety:
- Replace repeated terms with synonyms or alternative phrasing, e.g., use "intention to leave" instead of "quit intentions."
- Consistent Tense:
- Ensure uniform tense use throughout, adhering to academic conventions.
With these refinements, the manuscript can achieve high standards of academic English appropriate for publication.
